# Machine Learning based histology phenotyping to investigate the epidemiologic and genetic basis of adipocyte morphology and cardiometabolic traits

Craig A. Glastonbury [1,2‡*], Sara L. Pulit[1‡], Julius Honecker[3‡], Jenny C. Censin[1,4], Samantha Laber[1,5], Hanieh Yaghootkar[6,7], Nilufer Rahmioglu[4,8], Emilie Pastel[6], Katerina Kos[6], Andrew Pitt[9], Michelle Hudson[9], Christoffer Nellåker[1,8], Nicola L. Beer[10], Hans Hauner[3,11,12], Christian M. Becker[8], Krina T. Zondervan[4,8], Timothy M. Frayling[6,9‡], Melina Claussnitzer[5,13,14‡], Cecilia M. Lindgren[1,4,5‡*]

**1** Big Data Institute, University of Oxford, Oxford, United Kingdom, **2** BenevolentAI, London, United Kingdom, **3** Else Kröner-Fresenius-Center for Nutritional Medicine, School of Life Sciences, Technical University of Munich, Freising, Germany, **4** Wellcome Centre for Human Genetics (WCHG), Oxford, United Kingdom, **5** Broad Institute of MIT and Harvard, Cambridge Massachusetts, United States of America, **6** Genetics of Complex Traits, University of Exeter Medical School, Royal Devon & Exeter Hospital, Exeter, United Kingdom, **7** Research Centre for Optimal Health, School of Life Sciences, University of Westminster, London, United Kingdom, **8** Endometriosis CaRe Centre Oxford, Nuffield Department of Women's and Reproductive Health, University of Oxford, Oxford, United Kingdom, **9** NIHR Exeter Clinical Research Facility, University of Exeter Medical School, University of Exeter and Royal Devon and Exeter NHS Foundation Trust Exeter, United Kingdom, **10** Novo Nordisk Research Centre Oxford (NNRCO), Oxford, United Kingdom, **11** Institute of Nutritional Medicine, School of Medicine, Technical University of Munich, Munich, **12** German Center of Diabetes Research, Helmholtz Center Munich, Neuherberg, Germany, **13** University of Hohenheim, Stuttgart, Germany, **14** Beth Israel Deaconess Medical Center, Harvard Medical School, Boston, Massachusetts, United States of America

‡ CAG, SLP, and JH share first authorship on this work. TMF, MC, and CML are joint senior authors on this work.
* craig.glastonbury@benevolent.ai (CAG); celi@bdi.ox.ac.uk (CML)

**Data Availability Statement:** All Data, models and code are available through Github. https://github.com/GlastonburyC/Adipocyte-U-net.

## Abstract

Genetic studies have recently highlighted the importance of fat distribution, as well as overall adiposity, in the pathogenesis of obesity-associated diseases. Using a large study (n = 1,288) from 4 independent cohorts, we aimed to investigate the relationship between mean adipocyte area and obesity-related traits, and identify genetic factors associated with adipocyte cell size. To perform the first large-scale study of automatic adipocyte phenotyping using both histological and genetic data, we developed a deep learning-based method, the Adipocyte U-Net, to rapidly derive mean adipocyte area estimates from histology images. We validate our method using three state-of-the-art approaches; CellProfiler, Adiposoft and floating adipocytes fractions, all run blindly on two external cohorts. We observe high concordance between our method and the state-of-the-art approaches (Adipocyte U-net vs. CellProfiler: $R^2_{visceral} = 0.94$, $P < 2.2 \times 10^{-16}$, $R^2_{subcutaneous} = 0.91$, $P < 2.2 \times 10^{-16}$), and faster run times (10,000 images: 6mins vs 3.5hrs). We applied the Adipocyte U-Net to 4 cohorts with histology, genetic, and phenotypic data (total N = 820). After meta-analysis, we found that mean adipocyte area positively correlated with body mass index (BMI) ($P_{subq} = 8.13 \times 10^{-69}$, $\beta_{subq} = 0.45$; $P_{visc} = 2.5 \times 10^{-55}$, $\beta_{visc} = 0.49$; average $R^2$ across cohorts =

**Funding:** C.A.G received a pump priming grant from Novo Nordisk to carry out this work. The funders had no role in study design, data collection and analysis, decision to publish, or preparation of the manuscript.

**Competing interests:** The authors have declared that no competing interests exist.

0.49) and that adipocytes in subcutaneous depots are larger than their visceral counterparts ($P_{meta} = 9.8 \times 10^{-7}$). Lastly, we performed the largest GWAS and subsequent meta-analysis of mean adipocyte area and intra-individual adipocyte variation (N = 820). Despite having twice the number of samples than any similar study, we found no genome-wide significant associations, suggesting that larger sample sizes and a homogenous collection of adipose tissue are likely needed to identify robust genetic associations.

## Author summary

Fundamental aspects of biology such as how the size or number of adipocytes relates to obesity and cardiometabolic health are still unanswered. To answer such questions, fast, accurate and automated measurements need to be acquired, free from human biases. Glastonbury *et al.*, 2020 describe a novel machine learning method to perform rapid acquisition of adipocyte area estimates from histological imaging data. Using these imaging derived phenotypes, Glastonbury *et al.*, 2020 assess the relationship between adipocyte size and a range of cardio-metabolic comorbidities, demonstrating that adipocyte size can vary depending on where adipose is stored throughout the body. By tying genetics with imaging data, Glastonbury et al., 2020 were able to demonstrate that previous findings associating adipocyte size with Type 2 Diabetes variants, are likely to be false positives. This study provides a means of being able to scale up GWAS type analyses to imaging derived phenotypes.

## Introduction

Although obesity is a heritable and heterogeneous cardiometabolic risk factor, little is known about how genetic variation influences human adipocyte size across adipose depots or how such variability may confer risk to obesity and other cardiometabolic outcomes [1–4].

A defining feature of obesity is an excess of white adipose tissue (WAT). WAT mass expansion can occur in a range of adipose depots. The two most well defined depots are subcutaneous WAT and visceral WAT, where adipose accumulates in intra-abdominal depots present mainly in the mesentery and omentum and which drains through the portal circulation to the liver [5]. WAT expansion, both in normal development and in the development of obesity, is defined by two mechanisms: (i) *hyperplasia*, the increase in the number of adipocyte precursor cells, leading to an overall increase in the number of mature adipocytes; and (ii) *hypertrophy*, the increase in size of adipocytes due to lipid filling [6–8]. Reduced total adipocyte number has been associated with type 2 diabetes (T2D) [9], and increased adipocyte size has been associated with insulin resistance, dyslipidemia, hepatic steatosis, and the onset of T2D [10–11]. In addition, similar adipocyte sizes observed in BMI-concordant twins suggests a strong genetic background underlying adipocyte size [12]. To date, little is known about the genetic variation or molecular pathways that regulate adipocyte morphology (e.g., size, density, and morphology) [6,13], or how these link to biological mechanisms, whole-body obesity related traits such as BMI and waist-hip-ratio (WHR), and subsequent cardiometabolic disease [14].

We therefore sought to explore the relationship between mean adipocyte area and anthropometric traits like WHR and BMI, as well as investigate the genetic underpinnings of mean adipocyte area by combining histology data of fat tissue with accompanying genetic variation data from the same samples. Whilst adipocyte counting software exists [15], we chose to focus

on adipocyte area as a more tractable problem to solve. However, as a by-product of measuring adipocytes, we also get an approximate count (proportional to total fat mass).

To allow for rapid, automatic quantification and segmentation of mean adipocyte area in adipose histology slides from subcutaneous and visceral tissue collected from four independent research cohorts, we developed and applied a Convolutional Neural Network (CNN). For the first time to our knowledge, we couple the use of image-derived adipocyte area estimates to test for associations with BMI, WHR adjusted for BMI (WHRadjBMI), and a range of glycemic traits. Finally, we report the first genome-wide association study (GWAS) of adipocyte surface area to date, with the goal of identifying common genetic variants that associate with adipocyte morphology and to investigate previously published links to adipocyte morphology. Whilst several adipocyte measurement software exist [15–17], we demonstrate better accuracy and runtime.

## Results

### Applying a convolutional neural network to obtain region of interest proposals from thousands of histology slides and millions of cells

We ascertained histology and genotyping data from four independent cohorts (**Table 1**): (1) the Genotype-Tissue Expression (GTEx) Project, comprised of a multi-ancestry sample collected in the United States [18], with adipose tissue sampled from the lower leg (subcutaneous) and greater omentum (visceral); (2) the Endometriosis Oxford (ENDOX) project from the Endometriosis CaRe Centre, University of Oxford, with adipose tissue sampled from beneath peri-umbilical skin (subcutaneous) and from the bowel and omentum (visceral) of women undergoing laparoscopy for suspected endometriosis; (3) severely/morbidly obese patients undergoing elect abdominal laparoscopic surgery in the Munich Obesity BioBank (MOBB), with adipose tissue sampled from the upper abdominal area (subcutaneous) and the angle of His (visceral); and (4) a healthy cohort selected for not having type 2 diabetes (fatDIVA), with subcutaneous tissue sampled from the abdomen (see Methods for more detail).

To obtain adipocyte surface area measurements, we devised a deep learning pipeline that performs automatic classification of putative adipose cell containing Region of Interest (ROI) proposals in whole adipose tissue histology slides followed by segmentation of the images and then quantification, allowing us to filter tiles of slides that do not contain adipocytes (**Fig 1**).

Whole Slide Images are split into 1024 X 1024 pixel "tiles". A Convolutional Neural Network (CNN), InceptionV3, pretrained on ImageNet and fine-tuned on adipose histology tiles, is used to assign probabilities to tiles containing adipocytes. Using high confidence adipocyte containing tiles (*Posterior Probability* > 0.9) alongside manually created binary segmentation masks, we implemented a U-net CNN to segment adipocytes. We then apply a probability threshold to each segmentation probability map [19] (see **Data Availability and Code** for

**Table 1. Description of cohorts included in adipocyte morphology phenotyping and meta-analysis.** | Histology sample sizes denote the number of tissue samples available in either the subcutaneous (subq) or visceral (visc) depots, *after* image quality control was complete (see Methods).

| Cohort | N, histology (SC/VC)[†] | | % female | Mean age | Mean BMI | % with T2D | N, both genetic and histology data | |
|---|---|---|---|---|---|---|---|---|
| | *subq* | *visc* | | | | | *subq* | *visc* |
| GTEx | 715 | 562 | 34% | 53.4 | 27.5 | 22% | 504 | 410 |
| ENDOX | 308 | 42 | 100% | 32.9 | 26.5 | not available | 105 | 23 |
| MOBB | 142 | 171 | 67% | 46.5 | 44.4 | 30% | 113 | 131 |
| fatDIVA | 123 | 0 | 58% | 58.0 | 24.9 | 0% | 98 | 0 |

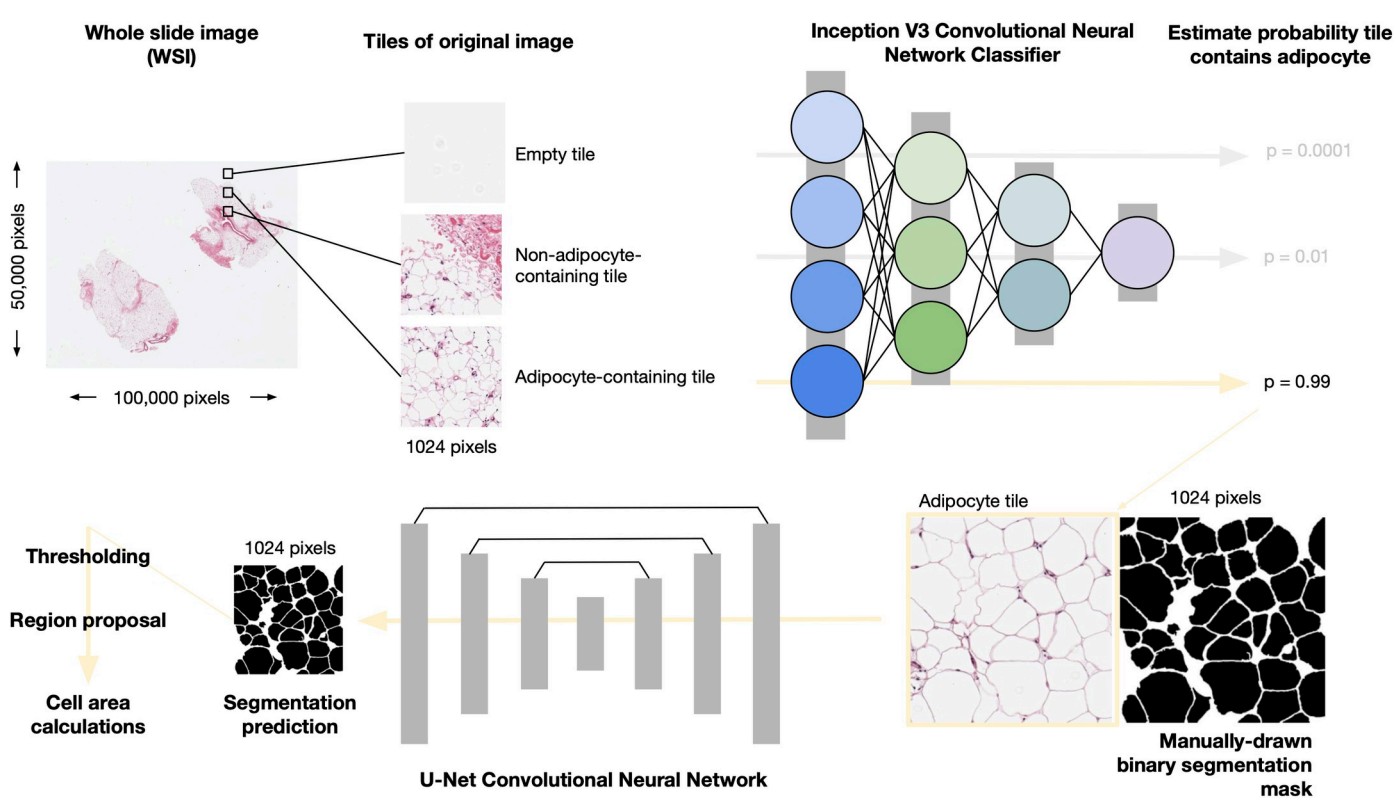

**Fig 1. Overview of the pipeline to obtain adipocyte areas.**

details), regions are proposed using scikit-learn and areas calculated. Pixel areas are converted into um$^2$ using the appropriate micron per pixel conversion factor per image.

First, we tiled each histology image slide using a sliding window of $1024 \times 1024$ pixels. We applied this tiling strategy across GTEx and ENDOX samples; the MOBB and fatDIVA cohorts already consisted of images containing adipocytes, and therefore required no tiling or filtering. Next, we manually selected tiles to form a training dataset of three distinct classes: (i) tiles containing adipocytes, (ii) tiles containing no adipocytes, and (iii) empty tiles. Example images of these three tile classes are shown in **Fig A in S1 Text**. To obtain only tiles containing adipocytes, we fine-tuned an InceptionV3 deep convolutional neural network (CNN) [20]; our CNN achieved 97% accuracy on our held out validation set (**Methods**). For each tile, we obtained the posterior probability that the tile belonged to one of the three defined tile classes (**Methods**) and defined the set of tiles containing adipocytes as those tiles exceeding a posterior probability threshold of $P > 0.90$ for being in that particular class. Choosing such a large posterior probability ensured we obtained images of just adipocytes and no other contaminant tissue (resulting in a low false positive rate and a high false negative rate). Examples of image tiles classified as adipocyte, non-adipocyte or empty at $P > 0.90$ are presented in **Fig B in S1 Text**.

### Using deep adipocyte U-net to robustly segment cells and estimate cell area

To measure adipocyte surface area, we treated the task as a segmentation problem. We created a training dataset of 175 high resolution, hand drawn, manually segmented (1024 pixels $\times$ 1024 pixels) binary segmentation masks across all four cohorts (**Methods**). Binary segmentation masks are images in which adipocyte/foreground are represented by white pixels taking on

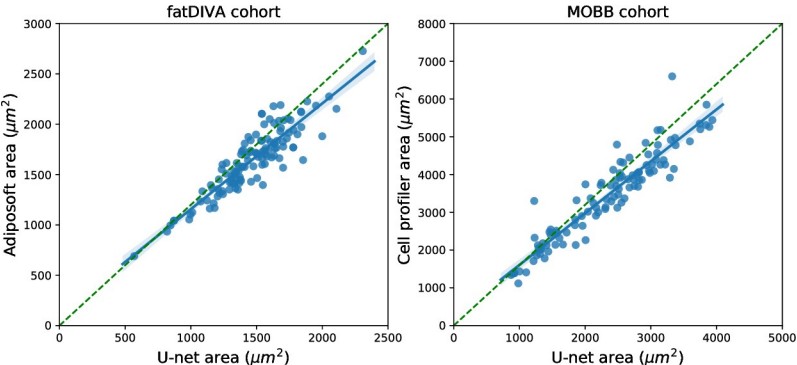

**Fig 2. Comparison of Adipocyte U-net-estimated adipocyte area with Adiposoft and CellProfiler-estimated adipocyte area across fatDIVA and MOBB cohort.** Estimates from our Adipocyte U-Net and from Adiposoft or CellProfiler are highly correlated, indicating concordance between our method and the two current gold standards for measuring cell morphology. The time necessary to compute these estimates with Adipocyte U-net was several orders of magnitude faster than the time required by Adiposoft/CellProfiler to generate the same measures.

value 1 and the background pixels are black, represented as 0 (normalised from pixel space [0–255] to [0–1]) (**Methods**). While automated segmentation methods such as watershed and adaptive thresholding methods (as implemented in Adiposoft and CellProfiler [17,21]) can be effective for some image analysis, deep learning has shown state-of-the-art performance in semantic segmentation, object recognition and biomedical segmentation tasks [22–25]. Additionally, our approach benefits from GPU-acceleration, as it is currently not feasible to analyse tens of hundreds to millions of images with traditional methods relying on serial CPU compute, Graphical User Interfaces (GUI), or both. Therefore, we trained a U-net architecture, which we call the Adipocyte U-net (**Methods**), to produce binary segmentation masks of adipocytes that are then trivial to count and measure computationally. Our Adipocyte U-net achieved a held-out performance dice coefficient of 0.84 (**Methods**), indicating a high degree of overlap between our predicted segmentation and the ground truth known segmentation in the heldout test set (see Methods).

To benchmark and validate our Adipocyte U-net, we used two cohorts that had previously independently (blindly) estimated adipocyte surface area for all individuals using Adiposoft (fatDIVA) and CellProfiler (MOBB), alongside significant manual gating and expert correction of area predictions. These two methods are the current state of the art approaches for segmenting both adipocyte histology images (Adiposoft) and images of a wider array of cells more generally (CellProfiler) [16,26,27]. We show significant concordance between independent adipocyte area estimates between either FatDIVA ($r_{subq} = 0.91$, $P = 8.3 \times 10^{-45}$) or MOBB ($r_{visc} = 0.94$, $P < 2.2 \times 10^{-16}$, $r_{subq} = 0.91$, $P\ 2.2 \times 10^{-16}$) and our novel adipocyte estimation method, Adipocyte U-net (**Fig 2**). Adipocyte area estimates from the Adipocyte U-net were, on average smaller, compared to the area estimates obtained by other methods. This can be attributed to a difference in cutoff values of cell size being used to exclude small, improperly gated objects between methods.

As a second, non-histology based validation strategy, we compared fat cell size from collagenase digestion to adipose U-net area estimates in the MOBB cohort. Independent of the depot, and similar to the histological validation above, we observed agreement between both methods ($n_{subq} = 46$, $r_{subq} = 0.41$, $P_{subq} = 5.0 \times 10^{-3}$; $n_{visc} = 65$, $r_{visc} = 0.59$, $P_{visc} = 2 \times 10^{-7}$) (**Fig O in S1 Text**). As previously reported, we find that whilst adipocyte area estimates vary substantially dependent on the method used for quantification, the correlation of adipocyte size and obesity is robust to these differences [28].

Our Adipocyte U-net required less than 6 minutes to predict adipocyte surface area across the ~10,000 images included in fatDIVA and MOBB. In comparison, Adiposoft took ~3.5 hours. GTEx and ENDOX, our largest cohorts, each consisting of approximately 250,000 images, took ~1 hour using the Adipocyte U-net, whilst we estimate Adiposoft would take 19 days for a single run. Additionally, our method captured adipocytes that are absent in Adiposoft produced segmentation masks (**Fig C** and **Fig D in S1 Text**). Examples of the test segmentation quality are presented in **Fig E in S1 Text** (all data and code available, see **Data Availability & Code**).

## Adipocyte area differences from adipose depots throughout the human body

We next utilised our Adipocyte U-net to obtain area estimates from our four cohorts. These cohorts totaled 2,176 samples (multiple distinct adipose depots per subject) (**Table 2**) making it the largest study of adipocyte morphology of its kind.

For adipocyte area estimation, we obtained estimates for the mean as well as the standard deviation of adipocyte size for 500 unique cells/per sample/per depot (subcutaneous and visceral). We determined 500 unique cells to be a necessary minimum for stable, low variance estimates of adipocyte surface area by applying Monte Carlo sampling (**Fig F in S1 Text**). Given the different metabolic and physiological roles subcutaneous and visceral adipose tissue depots play [29], we compared their mean adipocyte cell surface area and performed a random-effects meta analysis to compare adipocytes across visceral and subcutaneous adipose depots. A depot-specific effect was observed ($P_{meta}$ = 9.8 × 10$^{-7}$, β = -0.55), with larger cells on average observed in subcutaneous adipose depots, as previously reported [30] (**Fig G in S1 Text**). ENDOX cohort samples showed no significant, but directionally consistent, differences across the two depots (t-statistic = -1.52, $P$ = 0.13), likely due to limited power in this cohort (N = 42 visceral samples vs. N = 562 in GTEx; **Fig 3**). Finally, we observed variation within each depot, further demonstrating how adipocyte size within a single depot can vary substantially (**Fig 3** and **Fig I in S1 Text).**

As body fat distribution and it's genetic basis is sexually dimorphic [1,3], we tested for sexual dimorphic effects in adipocyte morphology. A depot-specific meta-analysis showed that mean adipocyte area in visceral, but not subcutaneous adipose, is sexually dimorphic (**Fig G in S1 Text**). Our meta-analysis indicated that women had smaller adipocytes in visceral fat ($P_{meta}$ = 3.05 × 10$^{-7}$, β = -0.34, I$^2$ = 0). While females have larger adipocytes in subcutaneous adipose as compared to men, this result was not significant when meta-analysed across cohorts ($P_{meta}$ = 0.08, β = 0.186 I$^2$ = 53.2), adjusting for BMI, age and ancestry (**Fig G** and **Fig J in S1 Text**). Due to the heterogeneity of subcutaneous adipose tissue being derived from various anatomical locations (I$^2$ = 53.2), it is possible there is a sexually dimorphic effect that is specific to precise anatomical subcutaneous adipose depots. For example, an effect (β = 0.32, $P$ = 3.0 × 10$^{-6}$)

**Table 2. Summary of adipocyte measurements per cohort.**

| Cohort | Mean adipocyte area estimates ($\mu m^2$) | |
|:---:|:---:|:---:|
| | Subcutaneous | Visceral |
| GTEx | 2,813 ± 717 | 2,352 ± 866 |
| ENDOX | 1,842 ± 484 | 1,711 ± 518 |
| MOBB | 3,239 ± 880 | 2,513 ± 850 |
| fatDIVA* | 1,461 ± 276 | N/A |

*The cohort fatDIVA were ascertained to fall within a healthy BMI range and to be free of type 2 diabetes. MOBB, with the largest cell size estimates, are primarily morbidly obese subjects.

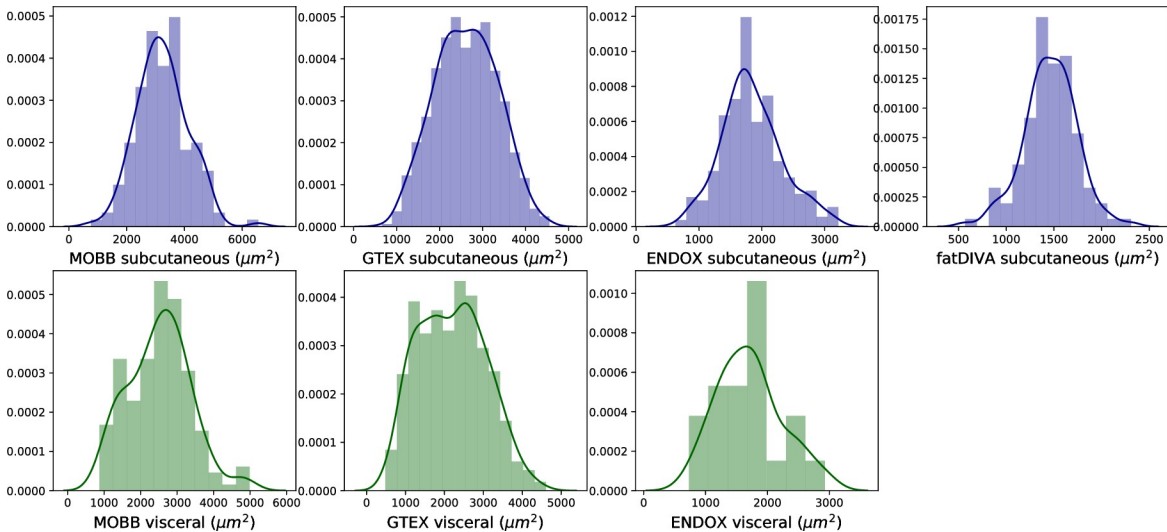

**Fig 3. Mean adipocyte area across adipose depots (subcutaneous: blue, visceral: green) per sample.** Visceral adipose tissue tends to have a bimodal distribution of mean adipocyte areas as compared to subcutaneous adipose tissue.

was present in GTEx (derived from the lower leg), one of the four cohorts analysed (**Fig G in S1 Text**).

Next, we assessed the relationship between adipocyte area in each depot and a range of disease relevant covariates. Previous studies have observed that obese individuals have larger adipocytes, but the vast majority of these analyses have been carried out using small sample sizes (N < 100) [30–31]. We recapitulate the relationship between adipocyte size and BMI in both subcutaneous and visceral depots across all four cohorts with an effective BMI range of 17–80, a range of collection methods and disease states. We observed an association between mean adipocyte area and BMI. We find that the mean adipocyte surface area in visceral fat correlates more strongly with BMI than adipocyte size in subcutaneous depots ($r_{subq} = 0.47$, $r_{visc} = 0.50$; **Fig 4**, $P_{meta} = 8.13 \times 10^{-69}$, $\beta = 0.45$), significant after adjustment for sex, age, T2D status, and

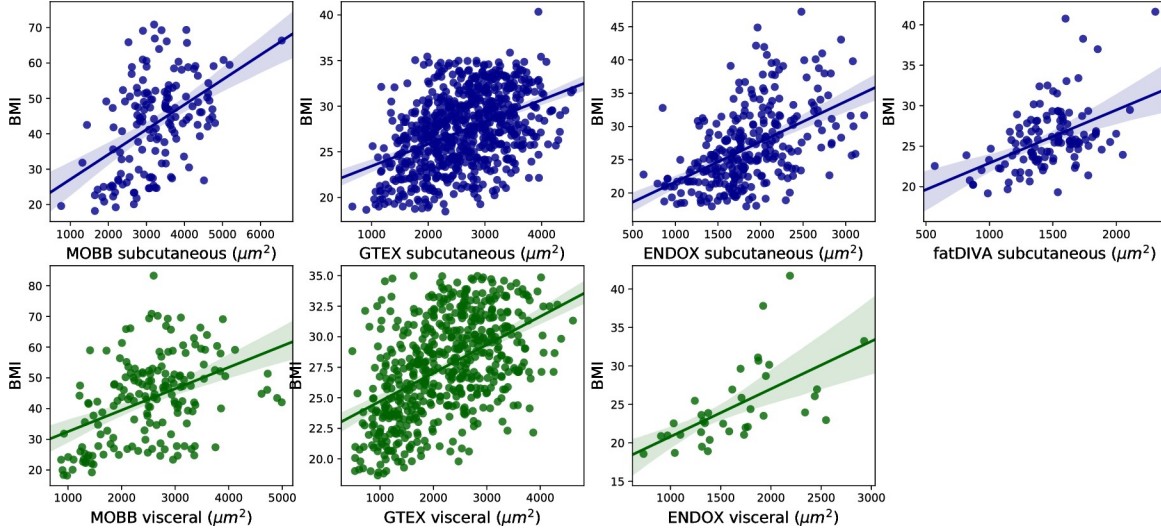

**Fig 4. Association between mean adipocyte area and BMI across subcutaneous and visceral adipose tissue depots.** We observe a strong correlation between BMI and mean adipocyte size across both subcutaneous and visceral depots in all cohorts.

self-reported ethnicity (**Fig G in S1 Text**). We also find a significant positive association between adipocyte area and subject age in visceral, but not subcutaneous adipose tissue, when meta-analysed across all available cohorts ($P_{subq\ meta}$ = 0.09, β = 0.09; $P_{visc\ meta}$ = 0.01, β = 0.12) (**Fig G in S1 Text**).

As GTEx samples are collected post-mortem, and numerous publications have shown a range of significant associations between sample ischemic time and assays performed on GTEx subjects [32–33], we assessed the relationship between our adipocyte area estimates, and sample ischemic time across depots. We find a positive association between mean adipocyte area and sample ischemic time ($P_{subq}$ = $1.7 \times 10^{-4}$, $β_{subq}$ = 0.14 ± 0.037; $P_{visc}$ = $6.8 \times 10^{-4}$, β = 0.14 ± 0.042) suggesting a relationship between longer ischemic time and larger cells, most likely due to cell degradation and/or bursting, leading to overestimation of cell surface area from broken or joining cell membranes or due to failed staining. Additionally, we found no association between cell estimates and self-reported ethnicity for both depots ($P$ = 0.59). Finally, studies have found conflicting evidence regarding a relationship (or lack thereof) between adipocyte size and insulin resistance or T2D status [31,34,35]. We meta-analyzed GTEx and MOBB, as T2D status was available for both, and did not observe a significant relationship between adipocyte size and T2D for either depot ($P_{visc\ meta}$ = 0.11, β = 0.12; $P_{subq\ meta}$ = 0.37, β = -0.19) (**Fig G** and **Fig K in S1 Text**). As a range of glycemic state variables ($HbA_{1C}$ and fasting blood glucose) were available for the MOBB cohort, we ran a multiple linear regression adjusting for BMI and age to investigate the relationship between adipocyte size and glucose homeostasis. Independent of the depot, no significant associations between glucose or $HbA_{1C}$ and adipocyte size were found (Glucose: $n_{Subq}$ = 110, $p_{Subq}$ = 0.71, $n_{visc}$ = 124, $p_{visc}$ = 0.061; HbA1c: $n_{Subq}$ = 79, $p_{Subq}$ = 0.75, $n_{visc}$ = 86, $p_{visc}$ = 0.059). While this suggests that BMI might act as the primary modifier of adipocyte size in these depots, further studies in larger cohorts including clinical biochemistry parameters of glucose homeostasis are necessary to clearly elucidate the role between diabetic state and adipocyte size.

For the MOBB cohort, we had additional extensive clinical measurements, including C-reactive protein, glucose, triglycerides and 12 additional clinical chemistry-derived phenotypes (**Fig H in S1 Text**). We observed several relationships between increased adipocyte size and WHR ($r_{subq}$ = 0.28, $r_{visc}$ = 0.33), C-reactive protein ($r_{subq}$ = 0.17, $r_{visc}$ = 0.26), prealbumin ($r_{visc}$ = 0.21), gGT ($r_{subq}$ = 0.17 $r_{visc}$ = 0.21), thyroxine T3 ($r_{visc}$ = 0.3) and triglycerides ($r_{visc}$ = 0.18), demonstrating a wide range of depot-specific associations. Of these associations, only BMI ($P_{subq}$ = $3.6 \times 10^{-10}$; $P_{visc}$ = $5.6 \times 10^{-11}$) remained significant after adjustment for multiple testing in 19 phenotypes ($P$ = $2.6 \times 10^{-3}$). Together, these analyses suggest that obesity, as measured through BMI, is the dominant phenotype associated with adipocyte size, with mean adipocyte size in visceral fat accounting for, on average, 25% of the variance of BMI (**Fig 4**).

## GWAS meta-analysis of depot-specific adipocyte surface area estimates

We sought to use our histology-derived phenotype data to identify common genetic variants (single nucleotide polymorphisms, SNPs) that associate with mean adipocyte size or variance of adipocyte size in subcutaneous and visceral tissue. After data quality control and imputation using the Haplotype Reference Consortium [36] for those cohorts with SNP array data (**Methods**), we used the histology phenotypes to perform genome-wide association testing in each depot and for each cohort. We then meta-analysed the results in an inverse variance fixed effects meta-analysis. In addition to performing the meta-analysis across the combined sample, we performed sex-specific analyses for each depot and each phenotype (mean adipocyte size and variance of adipocyte size).

Our meta-analysis in the combined sample in the subcutaneous depot examined 820 samples (424 women and 396 men) with available subcutaneous histology data, while our meta-

analysis in the visceral depot tested 564 samples (259 women and 305 men) with relevant available histology data. We performed all GWAS analyses using Plink 1.9 and adjusted analyses for sex, age, BMI, and the first 10 principal components (**Methods**). As the ENDOX cohort was genotyped on two separate platforms, we additionally adjusted these GWAS for genotyping platform. Due to ancestral heterogeneity in GTEx, we reduced our sample to just those individuals of European descent (**Methods**). We accounted for testing two separate depots by setting genome-wide significance at $P < 2.5 \times 10^{-8}$.

No SNP in our meta-analysis achieved genome-wide significance ($P < 2.5 \times 10^{-8}$), likely due to limited power given the small sample size. A small number of loci in our meta-analyses contained common SNPs at $P < 5 \times 10^{-7}$ with consistent direction of effect across all available cohorts (see summary-level results in **Data and Code Availability**), representing regions of interest for further genome-wide association testing.

Finally, we used our largest cohort (GTEx) to estimate the SNP-based heritability of the adipose histology phenotypes analysed here (**Fig L in S1 Text**). We used Genome-wide Complex Trait Analysis (GCTA) [37] to perform Restricted Maximum Likelihood (REML) analysis to estimate SNP-based heritability in each cell phenotype and across all sample groups (combined samples, women only, and men only). Cell phenotypes in both depots appear to be heritable traits, but error estimates were broad ($h^2$ of mean cell size in subcutaneous tissue = 35.3%, ± 38.4%; $h^2$ of mean cell size in visceral tissue = 22.4%, ± 48.5%; **Fig A in S1 Text**) reflecting that analysis of a larger set of samples is necessary for more accurate estimates of trait heritability.

## Association signals at previously published adipocyte size loci

Two adipocyte size common variant associations exist in the literature: the *KLF14* locus (rs4731702, found in 18 men and 18 women) and the *FTO* locus (rs1421085, found in 16 risk and 26 non risk-allele carriers). While the *KLF14* locus was characterised using adipose tissue histology from non-obese female individuals, the *FTO* locus was characterized using isolated floating mature adipocytes in subcutaneous adipose tissue from lean ($20 < BMI < 24$), population-level male individuals [38–39]. In our meta-analysis, we find no evidence to support associations for either rs4731702 ($P_{combined} = 0.925$, $P_{females} = 0.662$, and $P_{males} = 0.158$ for mean adipocyte size in subcutaneous tissue) or rs1421085 ($P_{combined} = 0.735$, $P_{females} = 0.426$, $P_{males} = 0.609$ for mean adipocyte size in subcutaneous tissue; **Fig B in S1 Text & S2 Table**).

*KLF14* is a female-specific type 2 diabetes-imprinted locus, it is only expressed from the maternally inherited allele. Additionally, rs4731702 has been linked to T2D risk in recent GWAS, with the locus only significant in female samples [38]. Because of this, we took further steps to mirror the original study design as best as possible in each of our cohorts. We excluded heterozygotes, only considering non-risk allele and risk-allele homozygotes, only considered genotype data from pre-menopausal women and subjects within a normal BMI range: $18 < BMI < 30$. After this stratification we observe a nominal association similar to the original study ($P = 0.012$ n = 18 risk allele carriers (CC) and n = 14 non-risk subjects (TT), having larger subcutaneous adipocytes (**Fig M in S1 Text**)). However, when we repeat this analysis in GTEx visceral fat samples and ENDOX subcutaneous fat samples, we see no evidence of association (GTEx, $P = 0.68$, $n_{CC}$ individuals = 9, $n_{TT}$ individuals = 9; ENDOX $P = 0.91$, $n_{CC} = 19$, $n_{TT} = 19$; **Fig N in S1 Text**). We were unable to perform these analyses in either fatDIVA or MOBB due to the absence of non-risk allele carriers remaining after sample filtering (fatDIVA) or due to the nature of the study design (MOBB, which contains a majority of morbidly obese subjects).

For *FTO* we were unable to similarly mirror the study design as we were able to do for the *KLF14* locus, given that the reported effect was specific to lean male individuals. Nevertheless,

we performed analysis to test for association between *FTO*-rs1421085 and adipocyte size in GTEx (n risk allele carriers (CC) = 73, n non-risk allele carriers (TT) = 246). In subcutaneous fat, *FTO*-rs1421085 was not significantly associated to adipocyte surface area in joint ($P = 0.39$, n = 319), female ($P = 0.75$, n = 125) or male-specific analysis ($P = 0.08$, n = 194), controlling for the effects of age and BMI. In visceral fat, *FTO*-rs1421085 was also not significantly associated to adipocyte surface area in joint ($P = 0.59$, n = 248), female ($P = 0.98$, n = 98) or male-specific analysis ($P = 0.34$, n = 150), controlling for age and BMI. Lastly, *FTO*-rs1421085 was not significant in ENDOX (female-specific subcutaneous cohort) ($P = 0.78$, N = 40), MOBB (morbidly obese) ($P_{subq} = 0.67$, $n_{subq} = 68$; $P_{visc} = 0.86$, $n_{visc} = 74$) or fatDIVA (normal range BMI and T2D free) ($P = 0.34$, n = 52).

## Association signals at previously published loci associated to obesity and fat distribution

We additionally sought to test whether known obesity and fat distribution loci were enriched for signal in our histology GWAS. We therefore looked up the index SNPs at loci associated to BMI and WHRadjBMI in our meta-analyses [3]. Of the 670 index SNPs associated to BMI and the 346 index SNPs associated to WHRadjBMI, approximately 3–7% of these SNPs achieved p < 0.05 in our histology GWAS. Similarly, of the index SNPs associated to either BMI or WHRadjBMI in sex-specifc analyses, 2–5% achieved p < 0.05 in our sex-specific histology GWAS. These results indicate no significant enrichment for signal at BMI- or WHRadjBMI-associated SNPs in our histology GWAS, consistent with the limited power we see in our overall analysis.

We additionally looked up those SNPs in our histology GWAS achieving a 'suggestive' p-value ($p < 5 \times 10^{-7}$) in large-scale meta-analysis of BMI or WHRadjBMI. Only one SNP, rs72811236, achieves nominal significance (p < 0.05) in a GWAS of BMI restricted to women only. This SNP may represent a bona fide association to adipocyte size, but additional samples are necessary to establish whether or not a true association exists in this region.

## Associations between obesity-trait genetic risk scores and adipocyte area

We tested for associations between genetic risk scores (GRSs) for BMI, WHR, and WHRadjBMI and mean adipocyte area in both subcutaneous and visceral fat depots (**Methods**). We observed a nominal ($P < 0.05$) association between the BMI GRS and subcutaneous mean adipocyte area. Each 1-unit higher BMI GRS (corresponding to a predicted 1-standard deviation higher BMI) was associated with 210 $\mu m^2$ (95% CI 23–397$\mu m^2$, $P = 0.03$) larger mean adipocyte area, with comparable results for standardized adipocyte area (**S2 Table** and **S3 Table**). However, the association did not surpass our Bonferroni correction threshold of $P < 0.008$ (adjusting for three obesity trait GRSs and two fat depots). We observed no other associations between the obesity-trait GRSs and mean adipocyte area, but the confidence intervals were large, suggesting that larger sample sizes are needed to reliably assess these relationships.

## Discussion

Imaging data provides a rich resource to perform rapid, accurate, and large-scale cellular phenotyping. Here, we developed the adipocyte U-net, an image segmentation machine learning model, to rapidly and accurately obtain measurements of adipocyte area from multiple human adipose depots across the human body. Whilst previous studies have used unsupervised learning methods to extract unknown cellular phenotypes and then performed GWAS on these latent representations [40], this study is the first to our knowledge that ties specific

(supervised) machine learned image phenotypes to genetic variants. We used these image-derived phenotypes to establish relationships between obesity, age, sex, T2D, and a range of clinical covariates. However, most associations we find are attenuated and no longer significant after conditioning on BMI, suggesting BMI is the primary driver of adipocyte size (r = 0.43–0.59 across cohorts and adipose depots). Using adipocyte surface area as a cellular phenotype, we performed the first GWAS of adipocyte surface area. Genome-wide association testing revealed no SNP exceeding genome-wide significance ($P < 2.5 \times 10^{-8}$ after multiple test correction). Heritability estimates also indicated that adipocyte area is likely heritable, but much larger sample sizes are required to obtain tight confidence bounds.

Our approach represents an additional opportunity for the application of machine learning in genomics. The Adipocyte U-net not only enables rapid phenotyping (our method is many orders of magnitude faster than current state-of-the-art approaches), but also demonstrates how genetic association studies could begin to examine endophenotypes, such as histology imaging, rather than clinically-measured phenotypes, such as BMI or waist-to-hip ratio. Being able to interrogate high-dimensional endophenotypes in a GWAS framework may yield a more rapid uncovering of genetic variants directly linked to the biological mechanisms that underpin clinically-measured outcomes. Many such methods to derive phenotypes from images are currently being developed [22,25,41,42].

A small number of studies have previously identified common genetic variants as associated to adipocyte morphology phenotypes. We report mixed replication results at both rs4731702 (at the *FTO* locus) and rs1421085 (at the *KLF14* locus). Our initial meta-analysis results indicate no evidence for association at either SNP (**S1 Table** & **S2 Table**). It is likely given the weakness of the initial results published and their lack of power (n < 50) that these loci do not reflect true adipocyte size associated loci. Additionally, whilst our study has variable sample ascertainment between cohorts (for example, ENDOX is an endometriosis cohort while GTEx are population-level ascertained postmortem samples), any single cohort described here is at least twice the size of the original publications. The potential signals at these loci will require further validation in much larger meta-analyses.

We have performed the largest study of automated histology measurements using a GWAS approach. Despite this being the largest study of its kind, our total sample size is < 1,000 samples and we find no genome-wide associated SNPs. Our findings suggest that larger samples will be necessary to uncover associated genetic variants and more accurately estimate heritability and polygenic risk of these phenotypes. Whilst we were underpowered to obtain meta-analysis heterogeneity statistics with tight confidence intervals, and by using random-effects to account for additional per-cohort variability, we do indeed see significant heterogeneity ($I^2$) for phenotype-adipocyte size analyses, likely reflecting the heterogeneous collection of subcutaneous and visceral adipose tissue depots across cohorts, including differences in anatomical locations from which tissues were collected. Larger, more homogeneous samples will be especially useful to investigate sex-specific effects [1,3]. Finally, our study focuses exclusively on samples of European-ancestry, a well-described bias in human genomics, [43–44] and studies in diverse ancestral populations will be necessary to fully understand the biology of adipocyte morphology and how this links to obesity, a condition that affects populations worldwide.

Adipose tissue represents quite a homogenous and therefore easy tissue for segmentation, with many other tissues, such as the placenta, being a much more complicated mix of cells and structures [25]. Adipocyte U-net could be rapidly adapted to other cell types using as few as 50 annotations and the principles of transfer learning [45]. We envisaged future work to take advantage of the recent successes of meta-learning, in which many similar tasks, such as classifying or segmenting cells across a range of image types, are solved at the same time and any future task can be adapted to work with very few gradient updates [46].

We have developed a method to enable rapid and accurate phenotyping from histology data, enabling integration of larger histology and GWAS datasets with highly-scalable computational phenotyping methods for future studies. Such an approach can accelerate the exploration of the genetic underpinnings of cell phenotypes or other endophenotypes measured via imaging data, thus paving the way for further insights into how genetic variation may contribute to adipocyte morphology and how these mechanisms may contribute to downstream cardiometabolic disease.

## Methods

### Data and code availability

Relevant code and data, including images and annotations can be found at the following GitHub repository: https://github.com/GlastonburyC/Adipocyte-U-net. Here, you can also find links to download the summary-results from our GWAS analyses.

GWAS summary statistics used for PRS:

https://github.com/lindgrengroup/fatdistnGWAS/tree/master/SuppTable1

### Cohort collection, curation, and quality control

**GTEx.**   The Genotype Tissue and Expression (GTEx) Project was initiated to measure gene expression and identify expression quantitative trait loci (eQTLs) in 53 tissues. The project has been previously described [18]. Briefly, samples were collected in the United States. The vast majority of samples were collected postmortem. Tissues were collected and stored according to a released protocol.

We obtained 722 subcutaneous and 567 visceral/omentum adipose tissue GTEx histology slides. All histology images were stored as whole-slide, high-resolution binary 'svs' files. All histology slides were obtained at scale 0.4942μm per pixel and were therefore comparable across samples. To obtain images that were of reasonable resolution for downstream processing and analysis, we tiled across each of the histology slides, producing $1024 \times 1024$ pixel tiles.

All samples had missingness < 5%. We excluded samples based on the suggested sample exclusions from GTEx. These samples include large chromosomal abnormalities (e.g., trisomies, large deletions) and mosaics. Principal component analysis (PCA) indicated the cohort to be a multi-ancestry cohort (including African-, East Asian-, and European-descent samples), to be expected given that samples were collected from many different locations in the United States. Within ancestral groups, no sample had an outlying inbreeding coefficient (defined as $\geq 6$ standard deviations from the coefficient distribution).

To clean SNPs, we split the samples (roughly) into subsamples of reasonably homogenous ancestral groups (for QC purposes only). We dropped all SNPs out of Hardy Weinberg equilibrium (HWE) with $P < 1 \times 10^{-6}$.

Because samples were sequenced on two sequencing platforms (HiSeq 2000, HiSeq X) we performed an association test between the SNPs on each platform and removed any SNPs with substantially different frequencies ($P < 5 \times 10^{-8}$). We ran the same association test, but this time checking for frequency differences by library preparation group and removed any associated SNP ($P < 5 \times 10^{-8}$).

After checking transition/transversion ratio by (i) site missingness, (ii) quality-by-depth (QD), and (iii) depth of coverage, we removed all sites with missingness > 0.5%, sites with QD < 5, and sites with total depth < 9000 or > 33,000. The final dataset included 635 samples and >35M genetic variants with a minor allele count > 1.

We assessed the association of adipocyte area to a range of whole-body traits that were available in the GTEx dataset (BMI, Weight, Ischemic time & Type 2 Diabetes status).

**Abdominal laparoscopy cohort—Munich Obesity BioBank / MOBB.** We obtained subcutaneous and visceral adipose tissue histology slides from a total of 188 morbidly obese male (35%) and female (65%) patients undergoing a range of abdominal laparoscopic surgeries (sleeve gastrectomy, fundoplication or appendectomy). The visceral adipose tissue is derived from the proximity of the angle of His and subcutaneous adipose tissue obtained from beneath the skin at the site of surgical incision. Images were acquired at 20× magnification with a micron per pixel value of 0.193μm/pixel. Collagenase digestion and size determination of mature adipocytes was performed as described previously [47]. All samples had genotypes called using the Illumina Global Screening beadchip array.

Collaborators from MOBB—the abdominal laparoscopy cohort, sent DNA extracted from 192 samples to the Oxford Genotyping Center for genotyping on the Infinium HTS assay on Global Screening Array bead-chips. Genotype QC was done using GenomeStudio and genotypes were converted into PLINK format for downstream analysis. We checked sample missingness but found no sample with missingness > 5%.

To perform the remaining sample quality control (QC) steps, we reduced the genotyping data down to a set of high-quality SNPs. These SNPs were:

a. Common (minor allele frequency > 10%)

b. Had missingness < 0.1%

c. Independent, pruned at a linkage disequilibrium ($r^2$) threshold of 0.2

d. Autosomal only

e. Outside the lactase locus (chr2), the major histocompatibility complex (MHC, chr6), and outside the inversions on chr8 and chr17.

f. In Hardy-Weinberg equilibrium ($P > 1 \times 10^{-3}$)

Relevant information, including code and region annotations, can be found in the GitHub repository provided in the Data and Code Availability section at the beginning of the **Methods**.

Using this high-quality set of ~65,000 SNPs, we checked samples for inbreeding and heterozygosity (—het in PLINK), but found no samples with excess homozygosity or heterozygosity (no sample >6 standard deviations from the mean). We also checked for relatedness (—genome in PLINK) and found one pair of samples to be identical; we kept the sample with the higher overall genotyping rate. Finally, we performed PCA using EIGENSTRAT and projected the samples onto data from HapMap3, which includes samples from 11 global populations. Six samples appeared to have some amount of non-European ancestral background, while the majority of samples appeared to be of European descent. We removed no samples at this step, selecting to adjust for principal components in genome-wide testing. However, adjustment for principal components failed to eliminate population stratification, and we therefore restricted to samples of European descent only, defined as samples falling within +/- 10 standard deviations of the first and second principal component values of the CEU (Northern and Western European-ancestry samples living in Utah) and TSI (Tuscans in Italy) samples included in the HapMap 3 dataset [48–49]. Finally, sex information was received after initial sample QC was complete. As a result, one sample with potentially mismatching sex information (comparing genotypes and phenotype information) was discovered after analyses were complete and therefore remained in the analysis.

Beginning with all SNPs available in the MOBB dataset (~800,000), we first removed all SNPs with missingness > 5% and out of HWE, $P < 1 \times 10^{-6}$. We also removed monomorphic

SNPs. Finally, we set heterozygous haploid sites to missing, in order to enable downstream imputation.

The final cleaned dataset included 190 samples and ~700,000 SNPs. We note that histology data was not available for all genotyped samples.

**fatDIVA.** "fatDIVA" (Function of Adipose Tissue for DIabetes VAriants) is a recruit-by-genotype study aiming to understand more about the mechanisms of differences in adipose tissue function. Research volunteers were identified by the NIHR Exeter Clinical Research Facility (Exeter CRF) and recruitment facilitated within the Exeter CRF. Before recruitment into fatDIVA, approximately 6,000 anonymised DNA samples from the Exeter 10,000 (EXTEND) study were genotyped on the Global Screening Array and imputed to the Haplotype Reference Consortium reference panel. A genetic risk score of 11 variants was then calculated for each individual and weighted by effects on fasting insulin. These 11 variants formed an early version of a "favourable adiposity" genetic score–where collectively the alleles associated with higher fat mass were associated with a favourable metabolic profile, and vice versa [50].

Individuals falling into the 5% lowest tail of the weighted genetic score were contacted and, if agreeing to take part in the study, matched to age (± 4 years), sex and BMI (± 1 unit) to an individual in the highest 20% of the weighted genetic score. Inclusion criteria were age 18–75 and exclusion criteria were: treated Diabetes (including insulin and GLP-1 analogues), history of bariatric surgery and recent significant weight loss/gain (± 3 kgs in the last 3 months); connective tissue disease, pregnancy and lactation, inflammatory or consuming conditions, and the following medications: prescribed glucose-lowering medication, lipid-lowering treatment (such as statins, fibrates or ezetimibe) or other medication that alters lipids (such as beta blockers and diuretics), oral/IV corticosteroid treatment or loop diuretics (furosemide, bumetanide), antiplatelet and anticoagulation medication, methotrexate. All participants were asked to refrain from strenuous exercise and from eating very fatty meals in the 48 hours prior to coming into the clinic, then fast overnight prior to attending a one-off morning visit at the Exeter CRF. A sample of abdominal fat was obtained by firstly injecting some local anaesthetic into an accessible area of the abdomen. Using a scalpel, a small incision (approx 2-3cm) was made to a depth of approx 15mm and a small (pea-sized) sample of fat removed. The wound was closed with simple sutures or steristrips. Part of the fat sample was stored in formalin for later H&E staining. For each individual, a H&E stained slide was examined under a microscope and ten photographs of different parts of the slide taken, with the operator choosing sections with a clear vision of adipocytes wherever possible. Adiposoft software was used to identify and quantify the area of adipocytes.

Samples had previously been imputed using the HRC panel and sent in best-guess genotype format.

Phenotypic and genetic sex information was consistent for all samples with available sex information. For all samples with sex information missing in the phenotype data, we used the genotypic sex to infer sex of the sample. A relatedness check found two pairs of related samples (pi-hat > 0.125). Because all samples were imputed (missingness is 0) we arbitrarily removed one sample from each pair. One sample had an inbreeding coefficient > 6 s.d from the mean of the inbreeding coefficient distribution, and was therefore removed. PCA indicated all samples to be of European descent.

We removed all monomorphic SNPs from the dataset and removed any SNP out of HWE ($P < 1 \times 10^{-6}$). The final dataset comprised 254 samples and > 8.8M SNPs.

**ENDOX Endometriosis case/control study.** Samples were genotyped at the Oxford genotyping center on two arrays: the Affymetrix Axiom (UK Biobank chip) (n = 56) and the Illumina Infinium Global Screening Array (n = 127).

Samples were cleaned in a manner identical to those samples in the abdominal laparoscopy (MOBB) and fatDIVA cohorts. No samples had missingness >5% and all samples were consistent in phenotypic and genotypic sex (all female). No sample was an outlier in the inbreeding check, and no pair of samples appeared to be related (pi-hat threshold of 0.125, equivalent to a cousin relationship). PCA using HapMap 3 data showed that all samples were of European descent.

SNPs were cleaned in a manner identical to those samples in the abdominal laparoscopy and fatDIVA cohorts. The final cleaned dataset included 127 samples and ~685,000 SNPs on the Illumina Array and 56 samples and 655,000 SNPs on the Affymetrix array.

For the genotyped cohorts without imputation data (ENDOX and MOBB) we performed imputation via the Michigan Imputation Server. We aligned SNPs to the positive strand, and then uploaded the data (in VCF format) to the server. We imputed the data with the Haplotype Reference Consortium (HRC) panel, to be consistent with the fatDIVA data which was already imputed with the HRC panel. We selected EAGLE as the phasing tool to phase the data. To impute chromosome X, we followed the server protocol for imputing this chromosome (including using SHAPEIT to perform the phasing step).

**Region of Interest proposal: InceptionV3 CNN.** We defined our Regions of Interest (ROIs) "adipocyte-only" training set tiles as having little to no vessels, smooth muscle or other tissue/contaminate present and that were composed of well shaped, viable, non-ruptured adipocytes that filled the majority of the tile (>80%). To automate this procedure, we trained an InceptionV3 deep convolutional neural network (CNN) architecture using transfer learning [20]. Whilst the original InceptionV3 network was trained on 1000 ImageNet classes [51], we only wanted to classify empty, adipocyte-only and non-adipocyte containing tiles. To do this, we removed the final dense layer and replaced it with an AveragePooling layer with (8,8) convolutions and a stride of 8. Our final layer consisted of a fully connected layer with outputs representing our three classes. We used a softmax activation to obtain posterior probabilities of any given tile belonging to one of our three classes. For a tile to be classified as containing adipocytes, we use only high confidence calls (Posterior Probability > 0.9). We used Stochastic Gradient Descent (SGD) with Nesterov momentum (0.9) and a learning rate of $1.0 \times 10^{-4}$. We trained the network on 2,729 tiles, approximately equally distributed across each class from both subcutaneous and visceral depots. We used a 80:20 train:validation split. The model reached a training accuracy of 95% and validation accuracy of 96.6%. Our trained classifier and weights are available to use in a Jupyter notebook (see: **Data and Code Availability)** [52].

**U-net architecture.** To obtain robust count and area estimates of adipocytes we used a deep convolutional neural network architecture based on a modified U-net, originally designed to perform biomedical image segmentation in a low sample size regime [23]. We used 175 manually created ground-truth segmentations of adipocyte tiles of resolution $1024 \times 1024$. We demonstrate our network learns the correct segmentation mask and predicts adipocytes that are commonly missed by Adiposoft (**S3 Fig**). Each adipocyte tile and corresponding mask were concatenated to create a large '*ensemble image*' which we then sampled $1024 \times 1024$ input images from. For validation, we used 10% of the data ensuring that the same image samples never overlapped for training and validation. Our loss function was a dice binary cross-entropy loss, and a dice coefficient metric was used to assess performance. The Dice coefficient measures the degree of overlap between two segmentations (A: ground truth, B: predicted) and takes on a value between 0 and 1, with 1 representing a perfect score:

$$Dice_{A,B} = \frac{2(A \cap B)}{(A + B)}$$

The final validation Dice coefficient was 0.844. As an output, we obtain a pixel-wise probability map per input image of the same dimensions with each pixel classified as an adipocyte or not. The trained U-net model architecture, corresponding weights, and jupyter notebooks are all publicly available (see: ***Data and code availability)***.

**Adipocyte area estimation.** To obtain robust adipocyte area estimates we utilised the output of our Adipocyte U-net. To further refine our predictions, we thresholded the probability maps and transformed them into grey-scale images. To obtain counts and area estimates for every cell in the image, we used the '*regprop*' function in the scikit-learn library. As a quality control step, we removed cell area estimates less than 200μm$^2$ and greater than 16,000μm$^2$, which typically represented cell debris and joined adipocytes where H&E staining had failed, respectively. As the quality of the histology slides varied significantly, our ROI method and use of a fixed number of cells per sample was essential to avoid any significant sample specific biases.

**Network training hardware and software specification.** U-net training took approximately 20 hours and the InceptionV3 fine tuning ran in under one hour. Inception tile classification for all samples took 13 hours (classification of more than 2 Million images) and Inference/Prediction on $1024 \times 1024$ images for the U-net took 22 hours (240,000 images). All models were implemented in Keras/Tensorflow. All networks were trained on a single server with a one Titan X pascal NVIDIA card, 12Gb of GPU memory and 64GB of RAM.

**Phenotype-Adipocyte size meta-analysis.** All phenotype-adipocyte meta-analyses were conducted using the R package 'meta' and 'metafor' [53]. As our cohorts come from heterogenous populations and both subcutaneous and visceral adipose depots are taken from various anatomical locations, we chose to use a random effects meta-analysis to capture the distributional differences in adipocyte size across cohorts. Whilst underpowered to estimate heterogeneity accurately, we calculated $I^2$, a statistic that quantifies the proportion of the variance in the meta-analysis attributable to heterogeneity.

**Genome-wide association testing and meta-analysis.** In each cohort, we implemented a genome-wide association study (GWAS) of the available histology phenotypes. The GWAS in the GTEx data was performed directly on the genotypes generated from whole-genome sequencing [18]. For the other three cohorts, we performed GWAS on the best-guess genotypes resulting from imputation with the Haplotype Reference Consortium (HRC) [36]. Imputation dosages were converted to best-guess genotypes using Plink 2.0 [54]. Due to limited data availability for the X chromosome, we restricted our GWAS to the autosomal chromosomes.

We performed GWAS in each cohort using linear regression implemented in Plink 2.0 (—glm). We adjusted all GWAS for sex, age, BMI and the top ten principal components calculated from common genetic variation in the cohort using a high-quality set of markers (see ***Cohort collection, curation and quality control***). For the imputed genotype cohorts, we restricted regression to those SNPs with an imputation quality (INFO) score > 0.3. We applied no minor allele frequency threshold at this step and opted instead to filter on allele frequency once the meta-analysis was complete.

After performing GWAS within each cohort for each group of samples (all samples, women only, men only) and each phenotype (mean cell size, cell size variance) we generated quantile-quantile (QQ) plots stratified by both imputation quality score and minor allele count (**Data and Code Availability** for details) to check for excessive genomic inflation in particular bins of SNPs. We observed no evidence for stratification in any GWAS, and therefore proceeded with meta-analysis.

Once GWAS within each cohort were complete, we performed an inverse-variance fixed effects meta-analysis for each phenotype (in the combined and sex-stratified samples). We

implemented the meta-analysis in METAL [55]. Once the meta-analyses were complete, we again plotted stratified QQ plots (**Data and Code Availability** for details) to check for evidence of population stratification or other sources of confounding. We set genome-wide significance at $P < 2.5 \times 10^{-8}$, reflecting a Bonferroni correction for testing tissue from two adipose depots. As adipocyte mean size and adipocyte variance are highly correlated to one another ($r_{subq} = 0.914$ and $r_{visc} = 0.963$), we did not count the two phenotypes as independent tests.

**Genetic risk scores for obesity-related traits and adipocyte area.** We constructed GRSs for BMI, WHR, and WHRadjBMI using independent ($r^2 < 0.05$) primary ("index", associated with each obesity trait $P < 5 \times 10^{-9}$) SNPs in the combined-sexes analyses in a recent GWAS [3] (see **data availability**). We excluded SNPs with duplicated positions, missingness $> 0.05$, HWE $P < 1 \times 10^{-6}$, and minor allele frequency $< 0.05$ in the imputed data, after filtering on imputation info $> 0.3$ in the imputed cohorts and restricting the GTEx cohort to those of European ancestry and excluding one individual due to relatedness. For these analyses, the individual in MOBB with potential sex mismatch between genotypic and phenotypic sex was removed. Only SNPs available in all cohorts after quality control was included, resulting in a final set of 530, 259, and 274 SNPs for BMI, WHR and WHRadjBMI, respectively. The SNPs were aligned so that the effect allele corresponded to the obesity-trait increasing allele. GRSs were then computed for each participant by taking the sum of the participant's obesity-increasing alleles weighted by the SNPs effect estimate, using plink v1.90b3 [56].

We then investigated associations with subcutaneous and visceral mean adipocyte area per 1-unit higher obesity GRS, corresponding to a predicted one standard deviation higher obesity trait, using linear regression in R version 3.4. [57]. All analyses were performed both with adipocyte area in $\mu m^2$ and in standard deviation units, computed through rank inverse normal transformation of the residuals and adjusting for any covariates at this stage. We adjusted for age, sex, and ten principal components, and with and without adjusting for BMI in the GTEx, MOBB, and fatDIVA cohorts. As we did not have access to data about age and BMI in the all-female ENDOX cohort, we only adjusted for ten principal components in that cohort and with and without adjusting for chip type. We then meta-analysed the cohorts, assuming a fixed-effects model. In the main meta-analysis model, ENDOX was included using the adjusted for chip type estimates. As a sensitivity analysis, we also reran the meta-analyses using the ENDOX estimates unadjusted for chip type and completely excluding the ENDOX cohort, yielding highly similar results.

## Supporting information

**S1 Text. Contains Fig A-O and Table A.**
(DOCX)

**S1 Table. Look-up of *KLF14* SNP reported to associate with adipocyte morphology phenotypes.** We looked up rs4731702, previously reported to be associated with adipocyte morphology phenotypes, in our own genome-wide association results. We find no evidence for association at this SNP (p > 0.05 for all phenotypes and all sample groups).
(TXT)

**S2 Table. Look-up of *FTO* SNP reported to associate with adipocyte morphology phenotypes.** We looked up rs1421085, a SNP previously reported to be associated with adipocyte morphology phenotypes, in our own genome-wide association results. We find no evidence for association at this SNP (p > 0.05 for all phenotypes and all sample groups).
(XLSX)

**S3 Table. Effect of obesity genetic risk scores on standardized and non-standardized adipocyte area in μm² in different fat depots.** Estimates derived from a linear regression model with estimates of adipocyte area in μm² per 1-unit higher genetic risk score, corresponding to a predicted 1 standard deviation higher obesity trait and adjusting for 10 principal components in all cohorts. In GTEx, fatDIVA and MOBB, adjustments were also made for age and sex, and with and without adjusting for BMI. In the all-female ENDOX cohort, adjustments were made for 10 principal components and with and without adjusting for chip type as the other covariates were unavailable. In the main meta-analysis model (assuming a fixed-effect model), ENDOX was included adjusted for chip type. Two sensitivity meta-analyses were performed; including ENDOX unadjusted for chip type and excluding the ENDOX cohort.
(XLSX)

## Acknowledgments

We thank study participants who donated adipose tissue to this study.

## Author Contributions

**Conceptualization:** Craig A. Glastonbury.

**Data curation:** Craig A. Glastonbury, Samantha Laber, Emilie Pastel.

**Formal analysis:** Craig A. Glastonbury, Sara L. Pulit, Julius Honecker.

**Funding acquisition:** Craig A. Glastonbury, Cecilia M. Lindgren.

**Investigation:** Craig A. Glastonbury.

**Methodology:** Craig A. Glastonbury.

**Project administration:** Craig A. Glastonbury, Nicola L. Beer.

**Resources:** Craig A. Glastonbury, Timothy M. Frayling.

**Software:** Craig A. Glastonbury.

**Supervision:** Christoffer Nellåker, Cecilia M. Lindgren.

**Validation:** Julius Honecker, Jenny C. Censin, Hanieh Yaghootkar, Nilufer Rahmioglu, Katerina Kos, Andrew Pitt, Michelle Hudson, Hans Hauner, Christian M. Becker, Krina T. Zondervan, Timothy M. Frayling, Melina Claussnitzer.

**Visualization:** Craig A. Glastonbury, Sara L. Pulit.

**Writing – original draft:** Craig A. Glastonbury.

**Writing – review & editing:** Craig A. Glastonbury, Sara L. Pulit, Julius Honecker, Cecilia M. Lindgren.

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
