## [Decision Letter · Decision Letter 0]

16 Jan 2020

Dear Dr Glastonbury,

Thank you very much for submitting your manuscript, 'Machine Learning based histology phenotyping to investigate the epidemiologic and genetic basis of adipocyte morphology and cardiometabolic traits', to PLOS Computational Biology. As with all papers submitted to the journal, yours was fully evaluated by the PLOS Computational Biology editorial team, and in this case, by independent peer reviewers. The reviewers appreciated the attention to an important topic but identified some aspects of the manuscript that should be improved.

We would therefore like to ask you to modify the manuscript according to the review recommendations before we can consider your manuscript for acceptance. Your revisions should address the specific points made by each reviewer and we encourage you to respond to particular issues, and specifically the lack of installation instructions and inability to reproduce the results noted by the reviewer #2. Please note while forming your response, if your article is accepted, you may have the opportunity to make the peer review history publicly available. The record will include editor decision letters (with reviews) and your responses to reviewer comments. If eligible, we will contact you to opt in or out.raised.

- Supporting Information uploaded as separate files, titled 'Dataset', 'Figure', 'Table', 'Text', 'Protocol', 'Audio', or 'Video'.

We hope to receive your revised manuscript within the next 30 days. If you anticipate any delay in its return, we ask that you let us know the expected resubmission date by email at ploscompbiol@plos.org.

Sincerely,

Lilia M. Iakoucheva, Ph.D.

Associate Editor

PLOS Computational Biology

Weixiong Zhang

Deputy Editor

PLOS Computational Biology

[LINK]

Reviewer's Responses to Questions

**Comments to the Authors:**

Reviewer #1: Glastonbury and colleagues present their results from an interesting study that ultimately did not deliver the desired genetic associations, but reflects an innovative approach and gives excellent insights into how more definitive studies should be performed.

Their approach is conceptually straightforward – they use novel analysis software to identify adipocytes in histological images, and then within the adipocyte component of the image the size of the adipocyte is measured. This new tool is called Adipocyte U-Net, and is shown to be computationally efficient. They then use this as a quantitative phenotypic trait and use genotypic information from the same patients to test for genetic associations with adipocyte size. The idea that we can harvest disease-relevant phenotypic information from histological images and study genetic associations is the most intriguing part of this report:

Author action item:

• Please add a short discussion somewhere whether this general approach (genetic association studies with quantitative data from histological images is completely novel or has some precedent.

The rationale for the study is that increased fat underlies increased body mass index (BMI), and that increased fat is due to either the hyperplasia (number of cells) or hypertrophy (size of cells) of adipocytes. Of these, it is only really practical to measure size, as it would be enormously challenging to count adipocytes accurately.

Author action item:

• Please discuss whether anyone has ever tried to measure adipocyte number, or even how in theory this might be performed.

Their Adipocyte U-Net approach is validated by a hold-out dataset, relative to other automated segmentation methods, and relative to data measuring single cell suspensions of adipocytes, performing well in each case.

The authors use samples of tissues from different parts of the body, and from four separate studies, finding that there are differences in sizes of adipocytes by anatomical site in the body, larger adipocytes in visceral fat in individuals with increased BMI, smaller adipocytes in visceral fat in women, but also larger adipocytes in samples with longer time in ischaemia.

Author action item:

• Where it states “Given the different metabolic and physiological roles subcutaneous and visceral adipose tissue depots play…” a citation would be valuable.

No association is found for adipocyte size and self-reported ethnicity.

Author action items:

• With genotyping data available, why was ancestry not assigned using genetic information? How variable were these subjects in terms of their ancestry?

• The causality statement “While this suggests that BMI might act as the primary modifier of adipocyte size…” is probably too assertive, as adipocyte size is later described to mediate BMI: “mean adipocyte size in visceral fat accounting for, on average, 25% of the variance of BMI”. These statements are a departure from the usual, more neutral descriptions in terms of associations, in which the direction of causality is not assumed.

The authors find no loci reaching significance for association with adipocyte size, but then focus on two loci believed to influence adipocyte size, KLF14 and FTO, not finding any evidence to support their roles, but expressing certain valuable caveats in reporting their results.

Author action item:

• Please explain what a “female-specific type 2 diabetes-imprinted locus” is. Uncited and unclear to a reader.

The major value of this paper is not in terms of novel insights, but in showing a path to a valuable approach to understanding the intermediate cellular phenotypes in human diseases. The Adipocyte U-Net tool is obviously cell type-specific, but it has to be assumed that other deep learning-based methods can be developed for other cell types.

Author action item:

• Please comment on how relatively easy or difficult it will be to develop comparable approaches that work for cells other than adipocytes. Are adipocytes relatively easy, will any cell types be predicted to be difficult, what sort of approaches would be need to count cell subtype proportions within a tissue or image area.

As such, this is a valuable and far-sighted report and should be in the public domain. The GitHub links work, demonstrating the commitment of the authors to sharing code and data.

Reviewer #2: Glastonbury et al. developed a deep-learning method called Adipocyte U-Net to rapidly estimate adipocyte areas from large-scale histology images and use them to study the relationship with obesity-related traits. Applying Adipocyte U-Net to four population-scale cohorts they demonstrate that adipocyte area positively correlates with BMI but GWAS meta-analysis shows no genome-wide significant results with N=820.

The text is well written, and the general presentation is very good. Given the amount of data we generate in daily clinical and scientific life and the enormous potential of machine learning models to make use of these data, the method presented here can be very useful to the scientific community.

The only major point I would like to raise is that the tool in the current state cannot be used by other scientists. Unfortunately, there is no instructions on how to implement, install and run Adipocyte U-Net on the github page. Many data sets and images are available but there is no explanatory text provided and at least for me there was no coherent way how these data should be used. The code is available in rather big .npy scripts, which I was not able to open with jupyter notebook. A user-friendly step-by-step manual would be highly advised. I truly believe that deep learning methods will advance science and medicine but if they are difficult to install and use the majority of users will swerve to tools that are not machine learning based just because of convenience.

Here is a list of comments to further improve this work:

- Please add 1-2 sentences in the introduction that list already available image processing tools that are suitable to phenotype adipocytes. A quick search showed AdipoCount (Zhi et al., 2018) and a newer Version of CellProfiler (McQuin et al., 2018), there might be more.

- According to CellProfiler 3.0 (“we’ve also made changes to CellProfiler’s underlying code to make it faster to run and easier to install, and we’ve added the ability to process images in the cloud and using neural networks“) it’s much faster now. I think it would be fair to compare run times using the latest version of CellProfiler. Please add those to the already existing run time comparison of Adipocyte U-Net vs Adiposoft and update corresponding text in the abstract if necessary.

- please provide a brief description of how Adiposoft (p. 22), CellProfiler (description is missing) and floating adipocyte fractions (p.20) quantifications were generated (e.g. software parameters, were all adipocyte fractions after collagenase treatment used or only certain fractions, etc.). This can be very brief but it would be nice to be able to roughly follow how other methods were used.

- is it known from the literature that adipocytes in subcutaneous depots are larger than their visceral counterparts or is this reported for the first time. Please specify.

- related to this question, can Adipocyte U-Net be applied to other tissues that are not predominantly fat tissue (e.g. arteries, muscles) but where biopsies are often “contaminated” or infiltrated with adipocytes. Do you expect adipocyte size differences in these different tissues?

- Often multiple biopsy samples are captured in the same image of a proband and some images show high inter-biopsy heterogeneity. Can Adipocyte U-Net be used to quantify the inter-biopsy heterogeneity to e.g. identify outlier images/proband or use it as an endophenotype itself?

- can the authors comment on the bimodal distribution seen in visceral tissues of Fig. 3? Did the authors check if this is a sampling issue or is there an underlying biological mechanism known? Providing example images of samples of each of the two peaks might be helpful.

- Did the authors check if known BMI GWAS hits replicate in their GWAS data of mean (or variance) adipocyte area? Or vice versa, do the suggestive signal of their GWAS replicate in published, highly powered BMI GWAS studies? The comparison of adipocyte size and BMI GWAS results would give further insights how endophenotypes and clinical phenotypes relate to each other.

- are the authors planning to make the individual-level mean and variance adipocyte size data available? I was not able to find them among the supplementary tables.

- out of curiosity, besides adipocyte size can Adipocyte U-Net be used to estimate the number of adipocytes of the whole image and use it as a phenotype for association with traits and genotypes? Or quantify the overall area of an image that contains adipocyte e.g. to automate the quantification of adipose tissue contamination/infiltration in e.g. arteries, muscles, etc?

- probably out of scope for this paper but did the authors check if mean adipocyte size (or its variance) is correlated with in silico adipocyte estimates measured by the authors themselves (Glastonbury et al. AJHG 2019) or available through the gtex portal (https://storage.googleapis.com/gtex_analysis_v8/interaction_qtl_data/GTEx_Analysis_v8_xCell_scores_7_celltypes.txt.gz, Kim-Hellmuth et al., biorxiv 2019)?

Minor points:

- When reading the abstract for the first time I was not sure what is meant by “adipocyte area” as it can be the size of an adipocyte but it can also be the area of an image that is occupied with adipocytes (similar to % fat tissue). It might help to use “mean adipocyte size” at the beginning or explain in a few words what is meant with adipocyte area.

- Fig. 2: Can the authors add a diagonal line to better visualize that Adipocyte U-Net tends to estimate smaller adipocyte sizes. If the MOBB cohort was analyzed using the old CellProfiler it should be rerun using CellProfiler 3.0 to compare Adipocyte U-Net with the latest versions of other methods.

- p. 7: please provide correlation plot that shows Adipocyte U-Net vs fat cell size from collagenase digestion.

- p. 9: Last sentence says “…one of the four cohorts analysed (Supplementary Figure 7)”, specifying the panel, I think it is panel B, would be good.

- p. 12: please provide correlation plot of mean adipocyte area and sample ischemic time.

- p. 20+21: text says “Using the remaining ~65,000 SNPs, we checked samples for inbreeding…” btu the last paragraph of the MOBB data set says “The final cleaned dataset included 190 samples and ~700,000 SNPs.” Is this a typo? If not, please add more details to this method section to clarify which step removes how many SNPs.

- please provide further statistics (boxplot, mean/median line, p-values of the group comparisons) mentioned in the text or legend for fig. S10, S11 and S13 for easier comparison. MOBB data was tested for the adipocyte area association with T2D but the corresponding plots are missing. Please add those as additional panels to fig. S11.

- Fig. S14: The text says “when we repeat this analysis in GTEx visceral fat samples and ENDOX subcutaneous fat samples, we see no evidence of association” however only ENDOX samples are shown in fig. S14. Unless fig. S13A shows GTEx visceral fat samples please provide the corresponding GTEx plot.

- “rs1558902.chr16.53803574.cohort-level.gwas.results” and the other three supplementary tables contain cohort names such as “frayling”, “ndog” and “julius”. Please specify which name belongs to which cohort in a separate tab or in the header. Or adjust names to match the ones used in the manuscript (Endox, mob, fatdiva).

- if the number of references is not limited I would suggest updating (or at least adding) the more recent citations (e.g. CellProfiler 3.0 McQuin et al., 2018 and GTEx most recent preprint (https://www.biorxiv.org/content/10.1101/787903v1).

**Have all data underlying the figures and results presented in the manuscript been provided?**

Reviewer #1: Yes

Reviewer #2: No: Installation instruction and user manual are not provided to the reviewer to test the software and reproduce results.

PLOS authors have the option to publish the peer review history of their article (what does this mean?). If published, this will include your full peer review and any attached files.

Reviewer #1: Yes: John M. Greally

Reviewer #2: No

Reviewer #3: No

---

## [Decision Letter · Decision Letter 1]

23 May 2020

Dear Dr Glastonbury,

Thank you very much for submitting your manuscript "Machine Learning based histology phenotyping to investigate the epidemiologic and genetic basis of adipocyte morphology and cardiometabolic traits" for consideration at PLOS Computational Biology. As with all papers reviewed by the journal, your manuscript was reviewed by members of the editorial board and by several independent reviewers. The reviewers appreciated the attention to an important topic. Based on the reviews, we are likely to accept this manuscript for publication, providing that you modify the manuscript according to the review recommendations. One of the reviewers noted that some of their comments were addressed in the rebuttal letter but no changes were made inside the revised manuscript - please address this in the revised manuscript.

Sincerely,

Lilia M. Iakoucheva, Ph.D.

Associate Editor

PLOS Computational Biology

Weixiong Zhang

Deputy Editor

PLOS Computational Biology

[LINK]

Reviewer's Responses to Questions

**Comments to the Authors:**

Reviewer #1: Uploaded as attachment.

Reviewer #2: My comments were all sufficiently addressed. I have no further remarks.

**Have all data underlying the figures and results presented in the manuscript been provided?**

Reviewer #1: Yes

Reviewer #2: Yes

PLOS authors have the option to publish the peer review history of their article (what does this mean?). If published, this will include your full peer review and any attached files.

Reviewer #1: Yes: John Greally

Reviewer #2: No
---

## [Decision Letter · Decision Letter 2]

11 Jun 2020

Dear Dr Glastonbury,

We are pleased to inform you that your manuscript 'Machine Learning based histology phenotyping to investigate the epidemiologic and genetic basis of adipocyte morphology and cardiometabolic traits' has been provisionally accepted for publication in PLOS Computational Biology.

Best regards,

Lilia M. Iakoucheva, Ph.D.

Associate Editor

PLOS Computational Biology

Weixiong Zhang

Deputy Editor

PLOS Computational Biology

Reviewer's Responses to Questions

**Comments to the Authors:**

Reviewer #1: The authors have addressed all of my concerns, I appreciate their responsiveness.

**Have all data underlying the figures and results presented in the manuscript been provided?**

Reviewer #1: Yes

PLOS authors have the option to publish the peer review history of their article (what does this mean?). If published, this will include your full peer review and any attached files.

Reviewer #1: Yes: John Greally

---

## [Editor Report · Acceptance letter]

13 Jul 2020

PCOMPBIOL-D-19-01963R2 

Machine Learning based histology phenotyping to investigate the epidemiologic and genetic basis of adipocyte morphology and cardiometabolic traits

Dear Dr Glastonbury,

I am pleased to inform you that your manuscript has been formally accepted for publication in PLOS Computational Biology. Your manuscript is now with our production department and you will be notified of the publication date in due course.

With kind regards,

Matt Lyles
